# Assessing Prevalence and Transmission Rates of Malaria through Simultaneous Profiling of Antibody Responses against *Plasmodium* and *Anopheles* Antigens

**DOI:** 10.3390/jcm11071839

**Published:** 2022-03-26

**Authors:** Sidhartha Chaudhury, Jessica S. Bolton, Leigh Anne Eller, Merlin Robb, Julie Ake, Viseth Ngauy, Jason A. Regules, Edwin Kamau, Elke S. Bergmann-Leitner

**Affiliations:** 1Center Enabling Capabilities, Walter Reed Army Institute of Research, Silver Spring, MD 20910, USA; sidhartha.chaudhury.mil@mail.mil; 2Biologics Research & Development, Walter Reed Army Institute of Research, Silver Spring, MD 20910, USA; jessica.s.bolton.ctr@mail.mil (J.S.B.); viseth.ngauy.mil@mail.mil (V.N.); jason.a.regules.mil@mail.mil (J.A.R.); 3Henry M. Jackson Foundation for the Advancement of Military Medicine, Bethesda, MD 20817, USA; leller@hivresearch.org (L.A.E.); mrobb@hjf.org (M.R.); 4U.S. Military HIV Research Program, Walter Reed Army Institute of Research, Silver Spring, MD 20910, USA; julie.a.ake.mil@mail.mil (J.A.); ekamau@mednet.ucla.edu (E.K.); 5Laboratory Medicine, Department of Pathology, David Geffen School of Medicine, University of California, Los Angeles, CA 90095, USA

**Keywords:** *Plasmodium*, mosquito saliva, antibody, serosurveillance, electro-chemiluminescence

## Abstract

Reliably assessing exposure to mosquitoes carrying malaria parasites continues to be a challenge due to the lack of reliable, highly sensitive diagnostics with high-throughput potential. Here, we describe an approach that meets these requirements by simultaneously measuring immune responses to both disease vector and pathogen, using an electro-chemiluminescence-based multiplex assay platform. While using the same logistical steps as a classic ELISA, this platform allows for the multiplexing of up to ten antigens in a single well. This simple, reproducible, quantitative readout reports the magnitude, incidence, and prevalence of malaria infections in residents of malaria-endemic areas. By reporting exposure to both insect vectors and pathogen, the approach also provides insights into the efficacy of drugs and/or other countermeasures deployed against insect vectors aimed at reducing or eliminating arthropod-borne diseases. The high throughput of the assay enables the quick and efficient screening of sera from individuals for exposure to *Plasmodium* even if they are taking drug prophylaxis. We applied this assay to samples collected from controlled malaria infection studies, as well as those collected in field studies in malaria-endemic regions in Uganda and Kenya. The assay was sensitive to vector exposure, malaria infection, and endemicity, demonstrating its potential for use in malaria serosurveillance.

## 1. Introduction

Climate change and the continued destruction of balanced ecosystems contribute to an increased prevalence of insect-borne diseases and require the continued development and monitoring of the efficacy of vector control strategies [1,2]. While there are ample entomological assessment tools, challenges remain in assessing: (1) the level of disease transmission in populations, (2) trends in transmission over long periods of time, (3) individuals and populations with recent exposures (within several months), (4) focal areas or populations with ongoing transmission, and (5) populations at high risk. The present study aims to profile vector- and disease-specific antibody responses (serosurveillance) in resident populations to identify serological biomarkers that reliably determine human–vector contacts. To date, serological surveillance approaches are based on detecting antibody responses to the mosquito saliva protein gSG6 that was originally identified in *Anopheles gambiae* [3]. This antigen is only expressed in adult female mosquitoes and is required for blood feeding. While gSG6 has been an accepted marker for exposure to *Anopheles gambiae*, less is known about its sensitivity and specificity in detecting exposure to other *Anopheles* species or even other mosquito vectors. While high antibody levels to gSG6 are associated with an increased risk of contracting insect-borne diseases [3], the presence of gSG6-specific antibodies does not directly determine exposure to a particular *Anopheles*-borne pathogen. Here, we address this challenge with a serological analysis approach that simultaneously measures markers of (1) exposure to vector (antibodies to mosquito saliva, which even allows identification of the insect species), (2) exposure to pathogen (platform can be adjusted or expanded to cover specific pathogens), (3) infection, and (4) transmissibility in the case of *Anopheles*-borne *Plasmodium falciparum* (*Pf*) malaria parasites.

The testing panel used for the present study consisted of the following analytes: two peptides representing the immunogenic regions of the mosquito saliva protein gSG6 (peptides 1, 2 [3]) to measure exposure to vector; two fragments of the pre-erythrocytic *Pf* antigen circumsporozoite protein (CSP); the central repeat region (NANP) [4] and the C-terminal region [5] of CSP, to measure parasite exposure; two antigens of the sexual blood-stage of *Pf,* merozoite surface protein (MSP)-1 and apical membrane antigen (AMA)-1 [6,7], to measure experience of blood-stage infection; and two *Pf* antigens expressed by sexual blood-stage of *Pf* (the parasitic form that is taken up by the mosquito during a blood meal), Pfs16 [8] and Pfs25 [9], to measure transmissibility. Notably, since malaria chemoprophylaxis inhibits blood-stage infection, exposure to *Pf* parasites should be detectable by antibody responses to pre-erythrocytic antigens, such as CSP, regardless of whether an individual is taking chemoprophylaxis.

We apply the panel to three types of blood samples in order to assess its utility for serosurveillance purposes. First, we validate the panel using blood samples collected from a controlled human malaria infection (CHMI) challenge, wherein subjects are deliberately infected by *Pf* in a controlled clinical environment via mosquito bite. In CHMI, malaria typically progresses through the liver stage (pre-erythrocytic) and into the blood stage, where it is detected and then treated. CHMI subjects will be exposed to vector antigens, as well as *Pf* antigens, through the asexual blood stage of infection, although this blood stage may be curtailed due to prompt testing and treatment. Second, we use samples collected from subjects following immunization using *Pf* irradiated whole sporozoite (IMRAS) [10]. These subjects receive IMRAS via mosquito bite, and the irradiated sporozoite is incapable of releasing infectious merozoites to start the blood-stage phase of the infection. As a result, IMRAS subjects will only be exposed to vector and pre-erythrocytic antigens. Finally, we apply the assay to test samples collected in moderate and high endemic regions of Kenya and Uganda that are at risk of natural infection. These individuals have likely been exposed to the vector, as well as all stages of malaria infection, including the sexual blood-stage, wherein the gametocytes responsible for transmission to mosquito are formed.

To date, multiplex serological assays have rarely been used for serosurveillance purposes despite their demonstrated value in serological assessments. In prior studies, we have demonstrated that multiplex serological assessment can be used to assess infectious disease exposure, vaccination, and correlates of immunity [5,11,12,13]. For example, in two recent studies on COVID-19, we were able to use this approach to characterize serological exposure in individuals with prior COVID-19 history and pre-pandemic samples to a range of coronaviruses, including SARS-CoV-2 [11,12]. We also used this approach to characterize the breadth of immunity induced by the RTS,S malaria vaccine [5]. While multiplex serological assays clearly have the potential to provide a rich assessment of immunity and exposure, more research and development, particularly in controlled laboratory settings, are needed. Here, we seek to develop a multiplex serological assay that simultaneously evaluates exposure to a vector and a vector-borne pathogen using *Anopheles* mosquito and malaria from samples collected both from controlled clinical studies, as well as in field studies in endemic regions. Successful development of this assay would provide a powerful new tool to complement existing entomological surveillance and gauge the endemicity and transmission of malaria in the field.

## 2. Materials and Methods

### 2.1. Samples

Malaria-naïve sera were obtained from healthy U.S. donors with no history of traveling to geographic regions with malaria through a blood collection protocol (WRAIR#2567) and a commercial source (Gemini Bio Products, West Sacramento, CA, USA) and served as negative controls (*n* = 25). Sera from individuals that underwent CHMI were collected under a clinical protocol (WRAIR#2572) (*n* = 9). Sera from individuals vaccinated with IMRAS were collected under a clinical protocol (www.clinicaltrials.gov, accessed on 22 March 2022, NCT01994525) (*n* = 21) [10]. The objective of this open-label clinical study was to determine the safety and identification of biomarkers of protection when exposing vaccinees repeatedly to bites from *Anopheles stephensi* mosquitoes as a means of vaccination [10]. Malaria-endemic samples were obtained from study-participants enrolled in a prospective study of acute HIV-1 infections in East Africa (Kericho, Kenya (*n* = 52, medium malaria risk) and Kampala, Uganda (*n* = 22, high malaria risk)), as previously reported [14].

### 2.2. Antigens

Recombinant *P. falciparum* (3D7 strain) proteins (MSP-1p42, AMA-1, Pfs25, Pfs16) were produced at Genscript (Piscataway, NJ, USA). Peptides derived from the *Anopheles gambiae* salivary gland protein (gSG6; peptides 1 and 2 [15]), circumsporozoite protein (CSP; representing the major repeat NANP [4], and the C-terminus [5] were synthesized by Atlantic Peptides). The antigens used as traps in the ECLIA assay were: gSG6-P1, gSG6-P2, CSP-NANP, CSP-Pf16 (*Pf* clone 3D7), AMA-1 (3D7), MSP-1 (*Pf* clone 3D7), Pfs16 (*Pf* clone 3D7), Pfs25 (*Pf* clone 3D7), and bovine serum albumin (BSA) as negative control.

### 2.3. Electro-Chemiluminescence Immunoassay (ECLIA)

The described multiplex ECLIA methodology is based on the Mesoscale U-PLEX platform utilizing 10-spot ECLIA plates (MSD, Gaithersburg, MD, USA) and performed as previously described [16]. Briefly, biotinylated proteins were diluted to a concentration of 300 nM using coating diluent (1× PBS with 0.5% BSA) and linked with a unique U-plex linker provided by the U-PLEX platform (MSD), vortexed, and incubated at room temperature (RT) for 30 min. The U-PLEX-coupled protein solutions were brought up to 6 mL with Stop Solution, creating a 1× multiplex coating solution. Plates were coated with the cocktail of proteins and incubated at RT for 1 h on a Titramax plate shaker (Heidolph, Schwabach, Germany), shaking at 700 rpm. Coated plates can be stored for up to seven days at 2–8 °C based on the manufacturer’s information. After incubation, the plates were washed with a working solution of 1× MSD Wash Buffer (MSD) three times. Sera were diluted to the desired concentration with Diluent 2 (MSD), added to each well, and incubated at RT for 1 h on a plate shaker. Plates were washed three times with 1× MSD Wash Buffer and incubated with the detection antibody, SULFO-TAG goat anti-human antibody (diluted to 1 µg/mL in Diluent 3 (MSD)). Plates were sealed and incubated at RT for 1 h on a plate shaker (700 rpm). After washing, MSD Read Buffer T was added to each well, and the plates were read on the MESO QuickPlex SQ 120 (MSD), per manufacturer’s instructions.

### 2.4. Statistical Analysis

The MSD assay provides a readout in units of mean luminescence intensity, and all readouts were directly log-transformed prior to analysis, without any normalization or subtraction of background. Univariate analysis comparisons between groups (geographic regions, malaria-endemic, vaccine recipients) were made using a Shapiro–Wilk normality test followed by a Student’s *t* test or a Wilcoxon signed rank test. We applied a multiple test correction using the Benjamin–Hochberg method; *p*-values were considered significant if their adjusted *p*-value was <0.05. Principal component analysis (PCA) was carried out by normalizing and scaling the log-transformed values. Data points were colored by group, and ellipses were generated corresponding to 50% confidence intervals for each group, to identify general trends in the data set. Correlation plots were generated using pairwise Pearson correlation coefficients calculated from the log-transformed data. All statistical analysis was carried out in R using the *stats*, *ggplot2*, and *corrplot*.

## 3. Results

### 3.1. Antigen Selection

We selected eight antigens for the panel that include *Anopheles* vector antigens (gSG6-P1 and gSG6-P2), two pre-erythrocytic antigens (CSP.NANP and CSP.Cterm), two asexual blood-stage antigens (MSP-1, AMA-1), and two sexual blood-stage antigens (Pfs16 and Pfs25). An overview of the different panel antigens within the context of the malaria transmission cycle is shown in Figure 1A. These antigens were selected for their ability to distinguish between vector exposure, parasite exposure, prior blood-stage infection, and transmissibility. We obtained samples from CHMI and IMRAS clinical studies to test the panel and then validated the panel using samples collected from field studies in malaria endemic regions of Kenya and Uganda. Figure 1B shows which stages of the transmission cycle individuals from these different studies would be expected to be exposed to. Because the CHMI study represents the closest surrogate of natural infection, albeit in a highly controlled environment, we used it to validate the assay in terms of identifying which antibody responses are induced following a single malaria infection episode.

### 3.2. Identification of Serological Markers of Malaria Exposure

Sera collected at day 9 and day 28 after study participants were exposed to vector bites in CHMI were tested in the Mesoscale platform against different antigens, as described in the Methods section. These antigens were multiplexed to establish a highly sensitive assay able to inform of exposure to both vector and pathogen. The results indicate that the repeat region of the CSP (NANP) is the strongest biomarker of exposure to *P. falciparum*-infected mosquitoes for this sample set, i.e., malaria-naïve individuals exposed once to *Plasmodium*-infected mosquitoes (Figure 2A). We observed induction of IgG antibodies specific to salivary protein gSG6, upregulated by a *Plasmodium* infection of the mosquito. We also saw an elevated antibody response to the *Plasmodium* blood-stage antigen MSP-1 compared to the negative controls. Future experiments will determine the longevity of antibody responses to the various antigens to assess whether the assay can distinguish between recent and past exposures.

Samples collected from subjects that underwent immunization with IMRAS via mosquito bite were also analyzed using the panel. Unlike CHMI, where the malaria parasite is able to develop into the blood-stage of infection, under IMRAS, the delivered parasite halts at the liver stage of development. We found that in the IMRAS samples only antibody responses to the repeat region of CSP were significantly different following immunization, showing that the panel is able to capture parasite development-stage specific differences in *Plasmodium* exposure (Figure 2B).

In summary, our assay platform enables the generation of serological profiles that inform about a recent malaria infection. The sensitivity of the assay is high enough to detect antibodies to mosquito saliva after a single exposure to *Plasmodium*-infected mosquitoes.

### 3.3. Characterization of Malaria Endemicity in Two Different Geographic Regions

Samples from malaria-endemic areas (Uganda, Kenya) were tested to establish antibody profiles of “malaria endemicity” in the malaria MSD panel to determine “malaria exposure” and “endemicity” in geographic regions with distinct malaria transmission rates (Figure 3). Due to the small sample size for CHMI subjects, we used IMRAS samples as a frame of reference for samples from a non-malaria endemic region to compare with samples collected from malaria-endemic regions.

We identified significant differences in the serological responses between the two geographic regions except for antibody responses to AMA-1, which was not different (summarized in Table 1). We also noted differences in the reactivity of the sera from Uganda vs. Kenya to the two peptides derived from the mosquito saliva protein gSG6 likely reflecting the predominance of different vectors at the two sites. Reactivity of antibodies to the two CSP fragments (i.e., CSP.NANP, CSP.Cterm) was higher in samples from Uganda, indicating higher endemicity and prevalence of malaria. Similarly, responses to MSP-1 were significantly higher (*p* < 0.001, *t*-test) in Uganda. In contrast, Pfs16 specific antibody responses were not different between the two regions. The degree of malaria endemicity is higher in Kampala/Uganda compared to Kericho/Kenya [17,18].

Serological responses were compared between the different cohorts (Uganda vs. Kenya, IMRAS vs. Kenya, IMRAS vs. Uganda) and malaria-immune (all three cohorts) vs. malaria-naïve samples (“neg”) (Table 1). Recent exposure to infectious bites (IMRAS) is associated with high antibody titers to CSP.NANP. Moreover, antibody titers to the mosquito saliva protein gSG6 are statistically different, compared to those in individuals not exposed to infected mosquitoes (“neg”). Based on the data, high transmission intensity (Uganda vs. Kenya) is associated with higher CSP and MSP-1 titers. Endemic exposure (Uganda/Kenya vs. neg) is associated with high levels of responses to all antigens.

Except for Uganda vs. Kenya, there were statistical differences in the antibody titers specific to the mosquito saliva, demonstrating the utility of these vector antigens as exposure markers.

### 3.4. Strong Correlations between Antigen-Specificities in the Serological Profile of Sera from Geographic Regions with Different Malaria Endemicity

After quantitatively assessing the antibody specificities, correlation matrices were generated that indicate which of the antigen-specificities correlate with each other, thus establishing a serological profile (Figure 4). Comparing the serological profiles of the two different geographic regions generated several key findings: (1) saliva gSG6 peptide 2 correlates with most of the malaria exposure markers including CSP (C-term), as well as the blood-stage antigens; (2) CSP-specific antibodies show geographic-specific correlation with other specificities: in Kericho (lower endemicity) repeat-specific antibodies do not correlate with the other responses; in Uganda (high endemicity), C-terminal antibodies do not correlate with the responses to the other analytes. These findings show that gSG6-specific responses are correlated with serological markers of malaria exposure and that the relationship between CSP responses and responses to other malaria antigens may be dependent on the level of endemicity or transmission intensity.

In conclusion, serological profiles have the potential to inform of malaria-endemicity; the fact that the saliva marker is strongly associated with exposure will provide invaluable information about the attack rate even in populations that are taking malaria prophylactic drugs.

### 3.5. Serological Profiles of Endemic Malaria Exposure

We used principal component analysis (PCA) to map the serological responses to vector and parasite between negative controls, IMRAS subjects, and samples collected from Kenya (low to moderate endemicity) and Uganda (high endemicity). We found that repeat-specific CSP responses were largely orthogonal to responses to the other vector and malaria antigens and formed a single axis that reflects recent malaria exposure and, indirectly, transmission intensity (Figure 5A). The responses to other vectors and malaria antigens form a second axis that likely reflects endemicity or experience to prior malaria infection. Here, IMRAS subjects show high responses along the CSP.NANP axis, reflecting parasite exposure, but low responses along the axis reflecting malaria endemicity. Samples from Uganda, show moderate to high responses along the CSP.NANP axis and moderate to high responses along the axis of malaria endemicity. Samples from Kenya are mixed—they show low to moderate CSP.NANP responses and moderate to high responses along the axis of endemicity.

These trends can be seen more clearly in a PCA using the three antigens that were found to be significantly different in the CHMI samples: gSG6-P1, reflecting the vector response; CSP.NANP, reflecting sporozoite exposure; and MSP-1, reflecting prior blood-stage infection (Figure 5B). Here, again, repeat-specific CSP responses form one axis, and responses to the vector antigen gSG6-P1 and the blood-stage antigen MSP1 form a second axis. IMRAS subjects who have repeated, high-level exposure to sporozoites shows very high CSP responses, with minimal gSG6-P1 and MSP-1 responses. Samples from both Kenya and Uganda show higher gSG6-P1 and MSP-1 responses, indicating prior exposure to both the vector and blood-stage infection., while Uganda samples show higher CSP responses, perhaps reflecting greater transmission intensity in that region.

## 4. Discussion

The present study establishes the utility and value of multiplexed, multi-antigen serology for entomological and parasitological surveillance. This tool is invaluable towards accomplishing malaria eradication, which requires the accurate assessment of pathogen prevalence and transmission rates. Conventional entomological surveillance, i.e., monitoring and evaluation methods have notable shortcomings: (1) labor intensive and costly; (2) highly variable, depending on trap type and abiotic factors; (3) vector trap sampling does not correlate with exposure to mosquito bites or disease—especially in the presence of interventions such as insecticide-treated clothing or bed nets; (4) sampling measures like human landing catches are associated with multiple concerns including extensive oversight requirements, intrinsic ethical issues, and operator biases.

Serological surveillance, i.e., measuring the presence and magnitude of antibodies specific to arthropod saliva and/or pathogens, has a direct relationship to exposure to vectors and the pathogen on both an individual and community level—indicating an evidence-based potential to serve as a proxy to or even replacement of conventional surveillance and evaluation methods [19,20,21]. Traditional serosurveillance methods rely on the ELISA assay, which has proven to be a robust, reliable tool for measuring antibody responses. However, newer multiplex assays such as electro-chemiluminescence platforms have been shown to have comparable ease of use and reliability compared to the ELISA, with greater throughput and higher sensitivity. In a head-to-head comparison of ELISA and ECLIA assays using the CSP antigen, we found that the multiplex ECLIA platform had a greater linear range and higher sensitivity, which permitted single-point measurements [22].

A recent study by Wakeman et al. applied a similar multiplex serology approach to measure antibody responses to vector and malaria antigens, using samples from Kisumu, Kenya, and from United States travelers with documented malaria infection, employing the Luminex platform [23]. Their 12-antigen panel was entirely peptide-based and included several antigens used in the present study including gSG6-P1, CSP, MSP-1, and AMA-1. There are some similarities between their findings and ours. For example, in endemic samples, they see high responses across virtually all antigens, compared to negative controls, including to the antigens listed above. They see a similar pattern in U.S. travelers with malaria, suggesting this broad seropositivity may result from immunity induced by even a single acute malaria episode, rather than (or in addition to) chronic or endemic exposure. However, there are some key differences to the present study. Their panel uses peptide-based antigens, which will likely not contain conformational epitopes found in recombinant antigens. They analyze pooled samples from Kenya, which masks individual variation in antibody responses. Furthermore, even though their panel included antigens from all stages of malaria, their samples were from individuals that had experienced the entire course of infection, and they do not report any *Plasmodial* life-stage specific results. By contrast, because we used samples from experimental models of infection, such as CHMI and IMRAS, that halt infection at specific life stages, we are able to demonstrate stage-specific serological patterns of malaria exposure. This has implications for serosurveillance under chemoprophylaxis, wherein malaria exposure may be specific to certain *Plasmodial* life-stages.

The current study demonstrates that the serological assessment of multiple antigens provides a more in-depth readout for exposure, malaria risk, and transmission rates than analysis of antibody responses to any single antigen. Antibodies to the mosquito saliva protein gSG6 are highly sensitive markers for exposure to arthropod vectors but only an indirect measure of risk for disease transmission. Simultaneously measuring pathogen-specific antibodies increases the accuracy of the assessment. In the CHMI and IMRAS samples, in terms of *Plasmodium* antigens, the main response was against the pre-erythrocytic CSP antigen, while in the CHMI samples there was some elevated response to MSP-1 but not to AMA-1, Pfs16, or Pfs25. This is to be expected in the IMRAS model that lacks the blood stage entirely, but it may be expected in the CHMI model as well, where blood-stage parasitemia is significantly curtailed by early (often pre-symptomatic) detection and rapid treatment.

The endemic samples showed significantly increased responses to all eight antigens compared to negative controls and to all antigens except CSP when compared to IMRAS. When comparing between Kenya (moderate transmission intensity) and Uganda (high transmission intensity), Ugandan samples had significantly higher responses to CSP and MSP-1 and a moderately higher response to AMA-1 but no significant difference in responses to the gametocyte antigens Pfs16 and Pfs25. The reason for this is unclear, and it may be that antibody responses to sporozoite and merozoite antigens are more sensitive to transmission intensity—perhaps because of their abundance and/or immunogenicity—compared to gametocyte antigens. Furthermore, antigen diversity could play a role as well. For example, the modest difference in AMA-1 that distinguishes samples from Kenya vs. Uganda may be due to the high polymorphic nature of the antigen [24,25], and only cross-reactive antibodies may be detected, thus limiting the sensitivity of the assay. It is important to note that, while these differences may be attributed to differences in transmission intensity [26,27], this study cannot distinguish whether this is due to the recency of the last malaria episode and/or the total number of malaria episodes in one’s lifetime.

Detection of mosquito-saliva specific responses in sera from CHMI participants demonstrated the potential of this readout in detecting exposure to the vector (Figure 1). We did not detect significant levels of gSG6-specific antibodies in IMRAS subjects despite the fact that study participants had received more than 900 infectious bites compared to participants in CHMI who received five infectious bites [10]. There are several possible reasons: (1) Technical issue (prozone effect [28]), i.e., interference in the detection of gSG6 specific signals due to excess antibody concentrations [28,29]). While this issue can be addressed by performing extensive dilution series, this was beyond the scope of the current study, as such extreme exposure rates are the exception; (2) Reports demonstrating that gSG6 specific antibodies are short-lived [30,31]; Post-immune sera collected 3–4 weeks after the last immunization fall within our established window for being able to measure these specificities; (3) Reports documenting that repeated exposure to gSG6 in mosquito saliva throughout the transmission season will lead to tolerance [30]; in this study, exposure to an immense dose of several thousand mosquito bites over a short period of time could possibly lead to the induction of tolerance, thus muting the antibody response against this vector-specific antigen. Future work will address the utility of other vector saliva antigens that are not subject to tolerance induction.

The current study focused on establishing antibody profiles from samples collected in African malaria-endemic regions with *P. falciparum* being the dominant parasite species and *Anopheles gambiae* [32,33,34] and *An. stephensi* [35,36] being prevalent vectors. Our assay panel could be adapted for analyses in other geographic regions where *P. vivax* and other mosquito species are prevalent by replacing *P. falciparum* antigens with comparable *P. vivax* antigens and by utilizing *Anopheles* species-specific homologs for gSG6. For example, reports from serosurveillance studies using ELISA-based detection of anti-gSG6 antibodies with samples from the South Pacific and in South America reported complete failure [37,38]. Moreover, the primary malaria vectors in Southeast Asia, *An. dirus*, *An. maculatus*, and *An. minimus*, only have a 48–87% identity with *An. gambiae* gSG6-P1 [39], and, therefore, it is likely that the antigen peptides used in our current panel will have to be replaced, resulting in geo-specific surveillance panels.

There are several shortcomings to the present study. First, while samples were collected from a longitudinal study conducted in endemic regions of Kenya and Uganda, no data on malaria history was collected from individuals from that study; thus, we could only assess the degree to which the assay was sensitive to malaria endemicity at a group level and not to malaria history at an individual level. Second, more data is needed on negative controls and samples collected from non-endemic regions in order to establish baselines and cutoff criteria for classifying subjects as exposed or quantifying the degree of exposure based on the serological readout. Third, more assessment is needed on the specificity of this assay, not just to different species of *Anopheles* mosquitoes, but also to different *Plasmodium* species such as *P. vivax,* which is predominant in south and southeast Asia. Fourth, the sample size is limited in this assay and may not capture the full breadth and diversity of serological responses in individuals living in the endemic regions of Kenya and Uganda. Further characterization of this assay using more samples from the broader region is necessary along with stratification of the results by demographic factors such as gender, age, and occupation. Finally, fifth, more development and testing of this assay is needed to ensure that it is robust against batch effects that might result from samples collected and stored under different conditions or for different periods of time.

## 5. Conclusions

The current study demonstrates the feasibility of a multiplex serological assay to determine exposure rates to mosquito vectors, pathogen, and disease. Furthermore, the data demonstrate the value of serosurveillance by establishing antibody profiles to analytes informing on exposure to vector, pathogen, and disease. Lastly, future studies will aim to address field performance of such assay platform with the goal having additional surveillance tools with higher throughput and accuracy available for field and clinical studies.

## Figures and Tables

**Figure 1 jcm-11-01839-f001:**
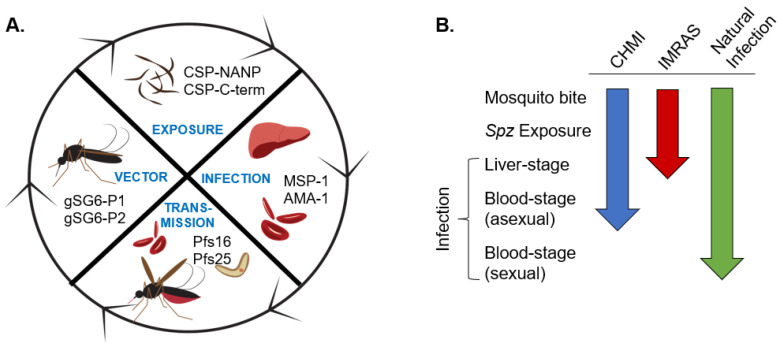
Antigen selection and malaria transmission cycle. (**A**) Antigens for vector, parasite exposure, infection, and transmission are shown along the malaria transmission cycle. (**B**) Different stages of exposure to the malaria transmission cycle for samples collected from individuals from CHMI, IMRAS, and natural infection in endemic areas.

**Figure 2 jcm-11-01839-f002:**
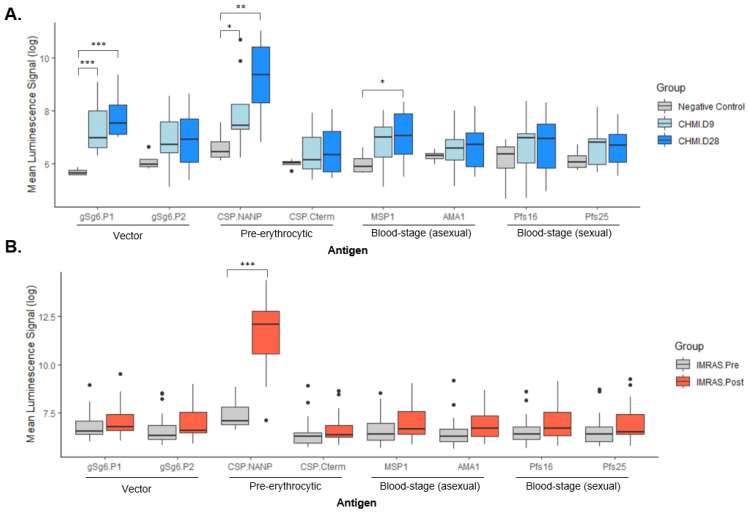
Specific antibody profiles after CHMI and IMRAS. (**A**) Malaria naïve volunteers were exposed to five infectious bites of *P. falciparum*-infected *Anopheles stephensi* mosquitoes in a controlled human malaria infection (CHMI, *n* = 12). All volunteers developed malaria by day 11 post exposure and required treatment with Malarone™. (**B**) Malaria naïve volunteers were immunized with irradiated whole sporozoites (IMRAS), and samples were collected before (pre) and after (post) immunization. All sera were tested at a 1:500 dilution. Asterisks indicate statistical significance (* *p* < 0.05, ** *p* < 0.01, *** *p* < 0.001). Black dots represent outliers.

**Figure 3 jcm-11-01839-f003:**
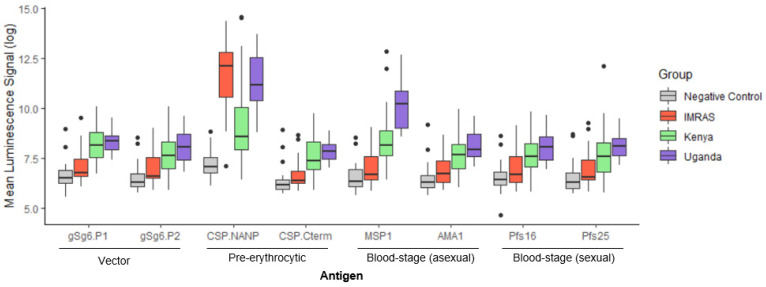
Specific antibody profiles from samples collected in endemic regions. The multiplex assay was tested against samples collected from Kericho, Kenya, from 2011–2014, which historically has low to moderate malaria-transmission intensity, and from Kampala, Uganda, from 2011–2018, which historically has high malaria-transmission intensity. Data from negative controls and the IMRAS samples are shown for comparison. Black dots represent outliers.

**Figure 4 jcm-11-01839-f004:**
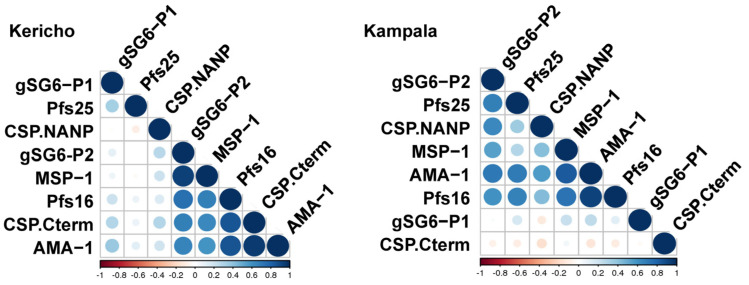
Correlation between antigen-specificities in the serological profile differ depending on malaria-endemicity. Correlation matrices depict the interactions between the different factors—the degree of correlation is indicated by color and color intensity (color bar indicates the legend for the statistical significance).

**Figure 5 jcm-11-01839-f005:**
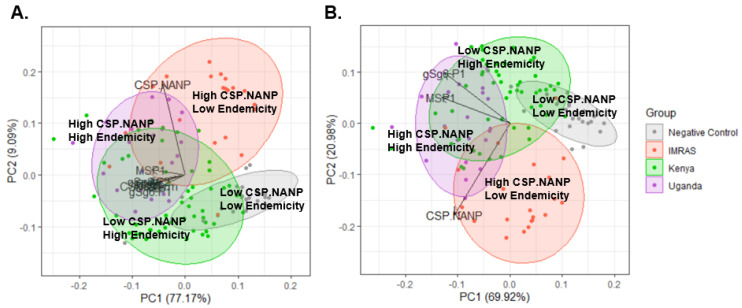
Serological profiles of endemic malaria exposure. (**A**) PCA using all antibody responses to all eight antigens for samples from negative controls (gray), IMRAS (red), Kenya (green, moderate transmission intensity), and Uganda (high transmission intensity). (**B**) PCA using antibody responses to a subset of three antigens that showed a significant difference following CHMI in clinical studies. Areas of the PCA plot that correspond to low and high CSP.NANP responses and low and high endemicity markers are shown.

**Table 1 jcm-11-01839-t001:** Statistically significant differences in antibody reactivities.

	Mosquito Saliva		*Plasmodium* Antigens					
			Pre-Erythrocytic		Blood-Stage (Asexual)		Blood Stage (Sexual)	
	gSG6-P1	gSG6-P2	NANP.CSP	CSP.Cterm	MSP-1	AMA-1	Pfs25	Pfs16
IMRAS vs. neg	0.042	0.114	<0.001	0.137	0.074	0.099	0.106	0.086
Uganda vs. Kenya	0.519	0.098	<0.001	0.165	<0.001	0.029	0.079	0.134
Kenya vs. neg	<0.001	<0.001	<0.001	<0.001	<0.001	<0.001	<0.001	<0.001
Uganda vs. neg	<0.001	<0.001	<0.001	<0.001	<0.001	<0.001	<0.001	<0.001
IMRAS vs. Kenya	<0.001	0.007	<0.001	<0.001	<0.001	<0.001	0.010	0.002
IMRAS vs. Uganda	<0.001	<0.001	0.448	<0.001	<0.001	<0.001	<0.001	<0.001

## Data Availability

All data are contained within the manuscript. R scripts can be obtained from the contributing author upon request.

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
