# Peer review of "Assessing Prevalence and Transmission Rates of Malaria through Simultaneous Profiling of Antibody Responses against Plasmodium and Anopheles Antigens"

_jcm, 2022, doi:10.3390/jcm11071839_

Round 1

Reviewer 1 Report

Chaudhury et al. JCMED

Overview:

The authors describing a multiplex approach enabling to measure the vector contact, the sporozoites injection, the development of blood stages and the differentiation in gametocytes over a single assay. The manuscript is well written and the point easy to understand. The convincing results demonstrate this strategy is working and I believe this approach could became a useful proxy to measure malaria incidence in various contexts and to complement measures only based on entomological metrics. However, this panel and the assay would need to be challenged in filed conditions with more accurate clinical information from the participants enrolled. This limitation is listed by the authors. I would have several comments. My main request would be that the authors discuss the added value of this method over what has been published before (see below).

comments:

Introduction section

First sentence, a reference would be welcome.
L50: does specificity isn’t more suitable than fidelity?

The paragraph L87 is not adding critical information for the manuscript, the use of serology to assess vaccination and pathogen exposure is to me evident

Results section

It would be interesting to see a statistical comparison in table 2 between CHMI D28 group and other groups.

Discussion section

A paper from Wakeman et al. JIM 2021 (https://doi.org/10.1016/j.jim.2021.113148) is proposing a similar approach, a comparison of the two studies would be interesting. Could the authors may insert this point in the discussion section.

The authors are using two cohorts from mid and high transmission area. This is confirmed with CSP and MSP1 but not Pfs25&16. We may assume that exposure to gametocyte was higher in the Uganda cohort but the difference looks minimal. How this dichotomy can be explained?

Line 303: the term ‘’far more sensitive’’ is to me not adapted, the multiplex approach is bringing the possibility to reach broader conclusion.

Line 309: antibody level against AMA-1 appear not conclusive in CHMI groups (D28 vs negative control). This should be discussed here as well. (fig2A)

Line 314: could you develop a bit more? Does the authors refer to PCR correction with the term marker of drug failure?

Line 335: please precise how this panel can be adapted to measure P. vivax incidence. Beside gsg6 which is likely to work, this would implicate to fully develop a new panel for parasites targets. Please precise these targets.

Author Response

COMMENT 1: Introduction: First sentence, a reference would be welcome.

>> We have added two references to support the statement

L50: does specificity isn’t more suitable than fidelity?

>> Both sensitivity and specificity are important, while the term ‘fidelity’ can be vague. We updated the relevant sentence:

“While gSG6 has been an accepted marker for exposure to Anopheles gambiae, less is known about its sensitivity and specificity in detecting exposure to other Anopheles species or even other mosquito vectors.”

COMMENT 2: The paragraph L87 is not adding critical information for the manuscript, the use of serology to assess vaccination and pathogen exposure is to me evident.

>> While the value of serology in assessing vaccination and exposure is self-evident, the utility of multiplex serology assays, particularly for the purposes of serosurveillance is an active area of study. We added a sentence to highlight the novelty of using multiplex serological assays for serosurveillance:

 “To date, multiplex serological assays have rarely been used for serosurveillance purposes despite their demonstrated value in serological assessments.”

COMMENT 3: Results: It would be interesting to see a statistical comparison in table 2 between CHMI D28 group and other groups.

>> Because of the small sample size of the CHMI study (n=9), we used it primarily to validate the serology assay. We chose to use the IMRAS samples (n = 22) as a frame of reference for malaria-exposed individuals from a non-endemic region, to compare with samples collected in malaria-endemic regions. We updated Paragraph 1 of Section 3.1:

“Because the CHMI study represents the closest surrogate of natural infection, albeit in a highly controlled environment, we used it to validate the assay in terms of identifying which antibody responses are induced following a single malaria infection episode.”

We also updated Paragraph 1 of Section 3.2:

“Due to the small sample size for CHMI subjects, we used IMRAS samples as a frame of reference for samples from a non-malaria endemic region to compare with samples collected from malaria-endemic regions.”

COMMENT 4: Discussion: A paper from Wakeman et al. JIM 2021 (https://doi.org/10.1016/j.jim.2021.113148) is proposing a similar approach, a comparison of the two studies would be interesting. Could the authors may insert this point in the discussion section.

>> We thank the reviewer for pointing out this very recent publication using the Luminex multiplex bead-based platform. Their approach is similar to our own in the use of an antigen panel that includes both vector and malaria antigens. However there are several key methodological differences. First, their 12-antigen panel is entirely peptide-based whereas our 8-antigen panel is largely based on recombinant proteins. Second, they use pooled samples from Kenya whereas we used individual samples. Pooled samples will mask individual variation, and likely bias the results towards seropositivity, because seropositivity of the pooled sample often results from the additive effects of seropositivity of each sample in the pool. Third, their samples come from individuals (or pools of individuals) that have experienced the entire course of malaria infection and they are not able to demonstrate stage-specific serological patterns of exposure. In addition to natural infection, we use samples from experimental models of infection, such as CHMI and IMRAS, which halt infection at specific Plasmodial life-stages.

We discuss some of the methodological similarities and differences and compare their results to our own in a new paragraph in the Discussion section:

“A recent study by Wakeman et al. used a similar multiplex serology approach to measure antibody responses to vector and malaria antigens, using samples from Kisumu, Kenya, and from United States travelers with documented malaria infection, using the Luminex platform [21]. Their 12-antigen panel was entirely peptide-based and included several antigens used in the present study including gSG6-p1, CSP, MSP1, and AMA1. There are some similarities between their findings and ours. For example, in endemic samples, they see high responses across virtually all antigens, compared to negative controls, including to the antigens listed above. They see a similar pattern with U.S. travelers with malaria, suggesting this broad seropositivity may result from immunity induced by even a single acute malaria episode, rather than (or in addition to) chronic or endemic exposure. However, there are some key differences with the present study. Their panel uses peptide-based antigens, which will likely not contain conformational epitopes found in recombinant antigens. They use pooled samples from Kenya, which will mask individual variation in antibody responses. Furthermore, even though their panel included antigens from all stages of malaria, their samples were from individuals that experienced the entire course of infection, and they do not observe any Plasmodial life-stage specific results. By contrast, because we used samples from experimental models of infection, such as CHMI and IMRAS, that halt infection at specific life stages, we are able to demonstrate stage-specific serological patterns of malaria exposure. This has implications for serosurveillance under chemoprophylaxis, where malaria exposure may be specific to certain Plasmodial life-stages.”

COMMENT 5: The authors are using two cohorts from mid and high transmission area. This is confirmed with CSP and MSP1 but not Pfs25&16. We may assume that exposure to gametocyte was higher in the Uganda cohort but the difference looks minimal. How this dichotomy can be explained?

>> We added a new paragraph to more fully explain how transmission intensity may be related to antibody responses to different stages of the parasite (Discussion, Paragraph 3):

“The endemic samples showed significantly increased responses to all eight antigens compared to negative controls, and to all antigens except CSP when compared to IMRAS. When comparing between Kenya (moderate transmission intensity) and Uganda (high transmission intensity), Uganda samples had significantly higher responses to CSP and MSP-1, a moderately higher response to AMA-1, but no significant difference in responses to the gametocyte antigens Pfs16 and Pfs25. The reason for this is unclear and it may be that antibody responses to sporozoite and merozoite antigens are more sensitive to trans-mission intensity – perhaps because of their abundance and/or immunogenicity – com-pared to gametocyte antigens. Furthermore, antigen diversity could play a role as well. For example, the modest difference in AMA-1 responses between Kenya vs. Uganda may be due to the high polymorphic nature of the antigen [20, 21] where only cross-reactive anti-bodies may be detected and thus limiting the sensitivity of the assay.”

 COMMENT 6: Line 303: the term ‘’far more sensitive’’ is to me not adapted, the multiplex approach is bringing the possibility to reach broader conclusion.

>> Agreed. The sentence has been revised:

“The current study demonstrates that the serological assessment of multiple antigens provides a more in-depth readout for exposure, malaria risk, and transmission rates than analysis of antibody responses to any single antigen.”

COMMENT 7:Line 309: antibody level against AMA-1 appear not conclusive in CHMI groups (D28 vs negative control). This should be discussed here as well. (fig2A)

>> While the CHMI model does have blood stage infection, it is significantly curtailed by early detection and rapid treatment, which may limit the antibody responses to AMA1. Paragraph 2 in the Discussion was revised to include the following:

“In the CHMI and IMRAS samples, in terms of Plasmodium antigens, the main response was in the pre-erythrocytic CSP antigen, while in the CHMI samples there was some elevated response to MSP-1, but not to AMA1, Pfs16 or Pfs25. This is to be expected in the IMRAS model that lacks the blood stage entirely, but it may be expected in the CHMI model as well, where blood-stage parasitemia is significantly curtailed by early (often pre-symptomatic) detection and rapid treatment.”

COMMENT 8: Line 314: could you develop a bit more? Does the authors refer to PCR correction with the term marker of drug failure?

>> We removed this sentence due to lack of clarity.

COMMENT 9: Line 335: please precise how this panel can be adapted to measure P. vivax incidence. Beside gsg6 which is likely to work, this would implicate to fully develop a new panel for parasites targets. Please precise these targets.

>> The reviewer is correct. We revised the sentence:

“Our assay panel could easily be adapted for analyses in other geographic regions where P. vivax and other mosquito species are prevalent by replacing P. falciparum antigens with comparable P. vivax antigens, and by utilizing Anopheles species-specific homologs for gSG6.”

Reviewer 2 Report

Questions:

  1. Why MSP1 values are not discussed in Figure 2A although they are significant. I understand that the authors just mentioned blood-stage infection in line 192 but this data needs to be discussed.
  2. The number of samples in the CHMI study is just 9. if possible more samples could be added to rule out the variations. The Standard deviation of CSP NANP CHMI D28 is so high. can this result be trusted as a low sample number is a major factor for this variance (significantly high value than other antigens)
  3. In figure 3: Why only IMRAS samples are included for comparison? Why not CHMI be added here to the table for comparison?
  4. Figure 3: The significant differences between the Kenya and Uganda samples for each antigen, could it be due to the differences in the individual samples when they are infected with the pathogen? These differences are also there in the gSg6 P1 and P2 although not significant. How can the authors clearly justify that Uganda's samples infection rate is high?

Minor:

Line 53: Change to does not directly

Line 127 to 138: Add space between values and units eg: 1 h, 300 nM 

Whole manuscript text: Correct spelling of mosquitos to mosquitoes

Line 262: Change to other vectors and malaria

Line 318: change to compared to 

Line 343: change to resulting in geo-specific

Figure 2: Make the Y-axis scales uniform (to log 15 in both A and B) which will be the same as figure 3. This is for the ease of the reader to evaluate the values.

Figure 5: The quality and resolution of the images are poor. please make high-resolution images

Author Response

COMMENT 1: Why MSP1 values are not discussed in Figure 2A although they are significant. I understand that the authors just mentioned blood-stage infection in line 192 but this data needs to be discussed.

>> Agreed. We added mention of this finding to Paragraph 1 of Section 3.2:

“We also saw an elevated antibody response to the Plasmodium blood-stage antigen MSP1 compared to the negative controls.”

We added further discussion of this in the Discussion (see response to Reviewer 1 Comment 7).

COMMENT 2: The number of samples in the CHMI study is just 9. if possible more samples could be added to rule out the variations. The Standard deviation of CSP NANP CHMI D28 is so high. can this result be trusted as a low sample number is a major factor for this variance (significantly high value than other antigens)

>> CHMI studies are typically carried out to assess the efficacy in vaccine clinical trials, and the unvaccinated ‘control’ cohort used to verify that the malaria challenge is working is typically a small group of 5-10 subjects. Because CHMI represents the closest surrogate to a natural infection, albeit in a controlled environment, we used it mainly to validate the assay. However, because of the low sample size, we did not include it in further analyses.

We updated Sections 3.1 and Section 3.2 to explain this in greater detail (see response to Reviewer 1 Comment 3).

COMMENT 3: In figure 3: Why only IMRAS samples are included for comparison? Why not CHMI be added here to the table for comparison?

>> We used only IMRAS samples for comparison because the small sample size of the CHMI study. The IMRAS samples served as a frame of reference for non-endemic samples to compare with the samples collected from endemic regions (Kenya and Uganda).

We added a sentence justifying the use of IMRAS for comparison in Section 3.2. See response to Reviewer 1 Comment 3 for further details.

COMMENT 4: Figure 3: The significant differences between the Kenya and Uganda samples for each antigen, could it be due to the differences in the individual samples when they are infected with the pathogen? These differences are also there in the gSg6 P1 and P2 although not significant. How can the authors clearly justify that Uganda’s samples infection rate is high?

>> Differences in antibody responses between Uganda and Kenya samples can be attributed to differences in transmission intensity, but in the present study we cannot distinguish between two aspects of higher transmission intensity: a more recent last infection or more total infections over one’s recent past or lifetime. We added the following to Paragraph 3 of the Discussion and references supporting our statement about the infection rate in Uganda vs Kericho/Kenya:

“It is important to note that while these differences may be attributed to differences in transmission intensity [24, 25], this study cannot distinguish whether this is due the recency of the last malaria episode and/or the total number of malaria episodes in one’s lifetime or recent past.”

MINOR COMMENTS:

Line 53: Change to does not directly. Done.

Line 127 to 138: Add space between values and units eg: 1 h, 300 nM. Done.

Whole manuscript text: Correct spelling of mosquitos to mosquitoes. Done.

Line 262: Change to other vectors and malaria. Done.

Line 318: change to compared to. Done.

Line 343: change to resulting in geo-specific. Done.

Figure 2: Make the Y-axis scales uniform (to log 15 in both A and B) which will be the same as figure 3. This is for the ease of the reader to evaluate the values. Done.

Figure 5: The quality and resolution of the images are poor. please make high-res images. Done.

Reviewer 3 Report

This is an intresting article on profilling on transmission rate against the antibody response.  
A few points of suggestion here. 

1. i think the keywords are too many. Usually we limit to 5 keywords to show that they are really significant.  
2. Page 2, line 65, typo (apical?) 
3. Elisa is a good diagnostic platform and versatile. More discussions are expected on technology platform and compare to others.

    https://doi.org/10.1101/721076
    https://www.nature.com/articles/s41467-021-21110-w
    https://www.nature.com/articles/nm.3622
    https://doi.org/10.1002/mrm.28387
    https://www.nature.com/articles/s42003-020-01262-z
    npj Aging and Mechanisms of Disease 6 (1), 1-12
4. Discussion about sampling size and how it affect the interpretation ?   

Author Response

COMMENT 1: I think the keywords are too many. Usually we limit to 5 keywords to show that they are really significant.  

>> We reduced the keywords to: Plasmodium, mosquito saliva, antibody, serosurveillance, electro-chemiluminescence

COMMENT 2: Elisa is a good diagnostic platform and versatile. More discussions are expected on technology platform and compare to others.

>> We agree with the reviewer that ELISA is a good and versatile diagnostic platform. Traditional serosurveillance is conducted using the ELISA. However, we have found that the multiplex electro-chemiluminescence assay (ECLIA) is superior in terms of ease of use, sensitivity, and throughput. We have elaborated on this in Paragraph 2 of the Discussion:

“Traditional serosurveillance methods rely on the ELISA assay which has proven to be a robust, reliable tool for measuring antibody responses. However, newer multiplex assays such as the ECLIA platform have been shown to have comparable ease of use and reliability of the ELISA with greater throughput and higher sensitivity. In a head-to-head comparison of ELISA and ECLIA assays using the CSP antigen, we found that the multiplex ECLIA platform had a greater linear range and higher sensitivity which permitted single-point measurements [20].”

COMMENT 3: Discussion about sampling size and how it affect the interpretation?   

>> We added a discussion of the sample size under the limitations of this study in Paragraph 7 of the Discussion:

“Fourth, the sample size is limited in this assay and may not capture the full breadth and diversity of serological responses in individuals living in the endemic regions of Kenya and Uganda. Further characterization of this assay using more samples from the broader region is necessary along with stratification of the results by demographic factors such as location, gender, age, and occupation.”

MINOR COMMENTS

Page 2, line 65, typo (apical?). It is not a typo: the apicoplast is an organelle found in the Plasmodium parasite. Apical refers to this organelle.